# Oral Health-Related Quality of Life in Different Subtypes of Ehlers-Danlos Syndrome

**DOI:** 10.3390/ijerph20032218

**Published:** 2023-01-26

**Authors:** Julius Balke, Lauren Bohner, Jeanette Köppe, Jochen Jackowski, Ole Oelerich, Marcel Hanisch

**Affiliations:** 1Department for Prosthodontics and Biomaterials, University Hospital Münster, D-48149 Münster, Germany; 2Department of Oral and Maxillofacial Surgery, Hospital University Münster, 48149 Münster, Germany; 3Institute of Biostatistics and Clinical Research, University of Münster, Schmeddingstraße 56, D-48149 Münster, Germany; 4Department of Oral Surgery and Policlinical Ambulance, Faculty of Health, Witten/Herdecke University, Alfred-Herrhausen-Str. 45, 58448 Witten, Germany

**Keywords:** Ehlers-Danlos Syndromes, EDS, rare disease, oral health, OHIP-14, oral health-related quality of life, OHRQoL

## Abstract

This study assessed differences in the oral health-related quality of life (OHRQoL) between subtypes of Ehlers-Danlos syndrome (EDS). For statistical analysis, participants were divided according to their subtype: classical EDS (cEDS), hypermobile EDS (hEDS), and vascular EDS (vEDS). All other subtypes were descriptively analyzed. Free-text questions and the German short form of the Oral Health Impact Profile (OHIP-14) were used. Finally, 295 questionnaires were included, representing 10 different EDS subtypes. The mean OHIP score of all participants was 19.6 points (standard derivation (SD) ± 12.3). The most predominant subtypes showed similar reduced OHRQoL, with 18.0 (cEDS, ±12.9), 19.5 (hEDS, ±12.0), and 15.2 (vEDS, ±11.6) OHIP points. For all other subtypes, the OHIP values varied. Participants waited an average of 21.8 years (±12.8) for their diagnosis. However, within the predominant subtypes, vEDS patients waited a noticeably shorter period of 13.3 years (±13.0; *p* = 0.004) compared to participants with hEDS. Additionally, this study showed no difference in OHRQoL for the predominant subtypes regardless of whether a participant was a self-help group member (18.8, ±12.0) or not (19.4, ±12.1; *p* = *0.327*).

## 1. Introduction

Ehlers-Danlos syndrome (EDS) is a clinically and genetically heterogeneous group of congenital connective tissue disorders [1], which is considered rare since it affects fewer than five in ten thousand people [2].

As of 2017, EDS can be categorized into 13 subtypes according to the international classification of EDS [1,3]. The classification of the 13 subtypes is based on different criteria; i.e., major and minor diagnostic criteria are given for each subtype. Major criteria either occur in almost all patients of the subtype or do not occur in other subtypes, and thus have a high diagnostic specificity. This allows the individual subtype to be distinguished from different subtypes or hereditary connective tissue disorders. Minor criteria, on the other hand, have lower diagnostic specificity and have supportive characteristics in diagnosis [3]. However, as the subtypes are subject to phenotypic variability and genetic heterogeneity [4], the definitive diagnosis is confirmed by molecular analysis of the genetic variant [3]. For all subtypes, except the hypermobile type (hEDS), molecular analysis is available. However, the genes responsible for the hypermobile subtype are still unknown [3].

At approximately 90%, the classical subtype (cEDS) and hEDS together make up the most common variants of EDS [5]. New data have shown that hEDS may not be rare and may affect more people than previously thought [6]. Other subtypes are rare and sometimes only occur a few times worldwide [3,7,8]. These include Arthrochalasia EDS (aEDS) with forty-nine confirmed cases, Dermatosparaxis EDS (dEDS) with fifteen confirmed cases, and cardiac-valvular EDS (cvEDS) with six confirmed cases, according to a study by Brady et al. from 2017 [7].

Due to mutations in the affected genes, defects occur in fibrillar collagen types I, III, and IV [1]. The genes can be either those encoding for the collagens or those encoding the enzymes involved in the post-translational modification [1]. These defects and mutations can result in various symptoms, such as hypermobile joints, skin hyperextensibility, and tissue fragility [3,7,8].

Since oral tissues are also affected by the collagen defects and mutations in EDS, various oral manifestations may occur. These include mucosal fragility, wound healing disorders, periodontal recession, and frequent temporomandibular joint dislocations [9,10,11,12,13]. These symptoms can influence oral health-related quality of life (OHRQoL). Previous studies have already evaluated the OHRQoL for EDS patients in general. For example, Berglund showed that EDS patients in Sweden had a reduced OHRQoL compared to the general population [14]. These results were consistent with studies by Hanisch et al. and Oelerich et al., who also found impaired OHRQoL in individuals with EDS [15,16]. However, no studies have analyzed how different EDS subtypes affect OHRQoL. Thus, this study evaluated how individual EDS subtypes can be affected by OHRQoL. Hence, the null hypothesis was given by:

H0: There is no significant difference in the Oral Health Impact Profile (OHIP-14) score between EDS subtypes.

## 2. Materials and Methods

The study was approved by the Ethics Commission of the Chamber of Dentists of Westphalia-Lippe and of the Westphalian Wilhelm University (2022-005-f-S). Data were collected from 1 February 2022 to 15 May 2022 using an online questionnaire in the German language.

### 2.1. Recruitment and Study Design

Electronic informed consent was obtained online before survey commencement by agreeing to a declaration of consent. To reach as many Ehlers-Danlos patients as possible, support groups from Germany (Ehlers-Danlos Selbsthilfe e.V. (www.bundesverband-eds.de, accessed on 22 July 2022), Deutsche Ehlers-Danlos Initiative e. V. (www.ehlers-danlos-initiative.de, accessed on 22 July 2022), Austria (SHG Ehlers-Danlos-Syndrom—Sozialinfo Wien), and Switzerland (Themenliste|Selbsthilfe Schweiz, accessed on 16 July 2022) were contacted via mail, with the help of which participants were informed about the study both by mail and via their website. In addition, social media groups that connect people with EDS were contacted, along with other researchers focused on EDS. In a previous study by Berglund et al., a mean OHIP-14 value of 11.1 across subtypes was observed, which was used as prior knowledge for sample size calculation [17]. It was determined that an effect could be demonstrated using a two-sided Kruskal-Wallis test with a power of 76.3% if data of N = 290 participants were available. The somewhat lower power used here was justified by the fact that the population under investigation represented a rare disease, which makes it challenging to recruit patients.

### 2.2. Eligibility Criteria

Participants had to be 18 years of age or older and a diagnosis of one of the Ehlers-Danlos subtypes according to the international classification of 2017 [3]. Only complete datasets were included. Misspellings and grammatical mistakes in free-form answers were no reason for exclusion if the overall message could be derived. Those cases with unclear answers were discussed by two authors (J.B. and O.O.) until consensus was achieved.

### 2.3. Data Collection

The questionnaire was generated with the help of Q-Set (https://www.q-set.de, accessed on 13 November 2022). Descriptive data were collected using free-text questions and a validated questionnaire in which the response could be selected. Subjective oral health-related quality of life was determined using the German short form of the OHIP-14 [18]. This questionnaire comprised 14 questions that dealt with topics such as phonetics, pain in the mouth, and satisfaction in everyday life related to food intake, teeth, mouth, etc. For each question, a score of 0 = ‘never’, 1 = ‘hardly ever’, 2 = ‘once in a while’, 3 = ‘often’ to 4 = ‘very often’ was assigned. Thus, the OHIP score took on values between 0–56 points, with 56 corresponding to the worst result. At the beginning of the questionnaire, participants were required to indicate their subtype of Ehlers-Danlos syndrome and were then accordingly divided.

### 2.4. Statistical Methods

To compare the means of the individual subtypes, t-tests at a significance level of α=5% were performed. In addition, effect sizes (Cohen’s d at a significance level of α = 5%) were calculated to determine the magnitude of the difference [19].

Based on the current recommendations, both the OHIP-G14 summary score and the four dimensions of OHRQoL (Oral Function, Orofacial Pain, Orofacial Appearance, and Psychosocial Impact) were calculated for all subtypes [20]. As recommended, only the scores of the OHIP-G14 domains “Physical Disability, Physical Pain, Psychological Discomfort, and Handicap” were calculated by adding the scores of the individual items (therefore, dimension scores could range from 0–8) [20]. Effect sizes for the four dimensions of OHRQoL were also calculated for the comparison of each subtype (Cohen’s d at a significance level of α = 5%).

A closed testing procedure was used to test the null hypothesis. Individual analysis was performed for cEDS, hEDS, and vEDS, because there were insufficient participants for the other subtypes. Therefore, the other seven subtypes were descriptively analyzed.

Using a two-sided Kruskal-Wallis test, the difference in the distribution of OHIP summary scores in cEDS, hEDS, and vEDS participants were first tested at a significance level of α = 5%. Then, if the null hypothesis could be rejected, all pairwise comparisons were made using a two-sided Mann-Whitney U test at a significance level of α = 5%. This analysis was also performed for the four dimensions of OHRQoL. For descriptive analysis, additional data were used as co-factors. Extracted data included:Primary outcome
OHIP score
Co-factors
General patient information
■Gender■Age■Country■Membership in a support group■ComorbiditiesDiagnosis
■Subtype■Age at the time of diagnosis■The time between the first symptoms and diagnosisOral health data
■The annual number of dentists visits■Involvement of the oral cavity■Number of lost teeth■Effectiveness of local anesthetics



*p*-values were calculated using a Chi-square test for categorical variables. In addition, a Kruskal-Wallis test was performed to calculate *p*-values for continuous variables. Finally, a multivariable linear regression analysis was performed to assess the impact of different co-factors on the primary outcome. Statistical analyses were performed using IBM SPSS Statistics for Mac, Version 28.0.1.0 (IBM Corp., Armonk, NY, USA), SAS software V9.4 (SAS Institute Inc., Cary, NC, USA), and RStudio Version 2022.07.1+554 (RStudio PBC, Boston, MA, USA).

## 3. Results

### 3.1. Participants

A total of 299 individuals participated in the study. Four patients were excluded because they were not eligible for this study. Two of those were excluded because they declined the declaration of consent. Another two were excluded because they were under 18 years of age.

Finally, 295 data sheets were included for assessment. Out of those 295 participants, 269 respondents lived in Germany. Eleven lived in Austria, and fifteen lived in Switzerland. A total of two hundred and seventy women (91.5%), nineteen men (6.4%), and six people with diverse gender (2.0%) participated in the survey. The mean age of participants was 39.2 years (range 18 to 65 years) at the time of the study. Detailed information can be found in Table 1.

### 3.2. EDS-Type

Of the thirteen known EDS subtypes, participants with ten subtypes took part in the survey. Of these, two hundred and thirty participants had been diagnosed with hypermobile EDS (hEDS, 78.0%), twenty-nine with classical EDS (9.8%), eighteen participants had vascular EDS (vEDS, 6.1%), seven participants had classical-like EDS (clEDS, 2.4%), three had periodontal EDS (pEDS, 1.0%), two had aEDS (0.7%), two participants had kyphoscoliotic EDS (kEDS, 0.7%), two had myopathic EDS (mEDS, 0.7%), one participant had cvEDS (0.3%), and one had Brittle Cornea Syndrome (BCS, 0.3%).

This study did not represent patients with dEDS, spondylodysplastic (spEDS), and musculocontractural EDS (mcEDS).

### 3.3. OHRQoL

The OHIP-14G over all subtypes had a mean score of 19.6 (±12.3) points in this study. The four dimensions of OHRQoL were calculated according to current recommendations, ranging from 0 to 8, with “Psychosocial Impact” having the lowest impact (1.92 ± 1.91) and “Orofacial Pain” having the highest impact on OHRQoL (4.26 ± 2.28). Detailed information regarding the four dimensions for each subtype can be seen in Table 2. Specific questions were often answered with a high score, equivalent to frequent occurrence. The inquiry about the frequency of pain in the mouth area was responded to by 132 participants with “often” or “very often” (44.7%). A total of 23 respondents reported that they “never” had pain in the mouth area (7.8%). Moreover, 109 participants (36.9%) stated that they “often” or “very often” had an unpleasant feeling when eating certain foods. A total of 69 participants (23.4%) stated that they “never” experienced an unpleasant feeling. One in three (100, 33.9%) respondents said they “often” or “very often” had difficulties relaxing or felt tense (98, 33.2%). A total of 104 participants reported an insecure feeling about their teeth, mouth, or dentures (35.3% often or very often). Detailed information for each OHIP-14 item can be seen in Appendix A.

### 3.4. Relationship between OHIP-14 and EDS Subtypes

No significant difference in OHIP score between the 10 subtypes participating in the study could be shown (*p* = 0.116). A sensitivity analysis was performed to evaluate whether individual subtypes significantly impacted the comparison between subtypes (Appendix A). No significant impact of a particular subtype could be shown. The answers to individual questions of the OHIP-14G questionnaire for each subtype can be found in Appendix A.

Because of the participation count, an additional analysis was performed for the three predominant subtypes. The mean score of OHIP in participants with cEDS was 18.0 (±12.9). Participants with hEDS had a mean score of 19.5 (±12.0), and the mean score of participants with vEDS was 15.2 OHIP points (±11.6). There was no significant difference in OHIP scores between these three predominant EDS subtypes (*p* = 0.327). Boxplots for each subtype can be seen in Figure 1.

In order to better compare individual subtypes and to investigate possible significant differences between individual subtypes, the mean differences and the related effect sizes were calculated (see Table 3). The one participant with BCS and the two participants with mEDS had notably higher OHIP scores compared to cEDS, vEDS, and hEDS participants. To compare the differences in the four-dimensional impact on OHRQoL between individual subtypes, effect sizes for each of the four dimensions were also calculated (Appendix A). Across all subtypes, a noticeable difference in the OHRQoL-dimension “Orofacial Pain” could be seen (*p* = 0.016). No noticeable difference could be seen for “Oral Function” (*p* = 0.090), “Orofacial Appearance” (*p* = 0.101), and “Psychosocial Impact” (*p* = 0.340).

### 3.5. Time of Diagnosis and Involvement of the Oral Cavity

At the time of diagnosis, the mean age of participants was 34.4 years (±12.3, range 1 year to 61 years). The mean time between the onset of first symptoms and diagnosis was 21.8 years (±12.8, range zero to 55 years). A total of 276 participants (93.6%) reported oral cavity involvement. An amount of 144 of the 295 respondents (48.8%) said that anesthesia had no effect or a decreased effect on them. Detailed information can be found in Table 4.

Across all subtypes, no noticeable difference in the time from the onset of the first symptoms to diagnosis could be seen using a Kruskal-Wallis test (*p* = 0.058). Again, additional analysis for the three predominant subtypes was performed (cEDS, vEDS, hEDS), in this case showing a noticeable difference in the time from onset of the first symptoms to diagnosis could be demonstrated (*p* = 0.003), with a pairwise comparison showing a noticeable difference between vEDS (13.3 years) and hEDS participants (22.5 years) (*p* = 0.004). Figure 2 shows the time between first symptoms and diagnosis for all subtypes.

### 3.6. Membership of a Support Group

One hundred fifty-six participants (52.9%) were organized in a support group. No differences in the OHIP between support group members were observed (*p* = 0.358).

The mean OHIP score of respondents, who were members of a support group or regularly attended support group meetings, was 19.0 (±12.3); for those not organized in a support group, the mean score was 20.2 (±12.4).

### 3.7. Is There a Difference in the Frequency of Dental Visits between Subtypes?

Proportionately, 6.9% of the cEDS patients, 10.0% of the hEDS patients, and 11.1% of the vEDS stated that they had not been to the dentist once in the last twelve months; while 55.2% of cEDS patients, 48.7% of hEDS patients, and 55.6% of vEDS patients went to the dentist once or twice in the past year. That they had visited the dentist more than twice in the last year was said by 37.9% of cEDS patients; 41.3% of hEDS patients and 33.3% of vEDS patients said the same.

### 3.8. Multivariate Analysis

The multivariable linear regression analysis showed noticeable effects on OHRQoL for diverse individuals (*p* = 0.005), age (*p* = 0.027), other comorbidities (*p* = 0.008), time between first symptoms and diagnosis (p = 0.024), involvement of the oral cavity (*p* = 0.01), and lost teeth (*p* = 0.006). Detailed information for all analyzed co-variables is shown in Figure 3.

## 4. Discussion

Comparing the OHIP scores among the predominant subtypes showed that the cEDS, hEDS, and vEDS participants had similar OHIP scores of 18.0, 19.5, and 15.2 points, respectively. Because of the low numbers of participants from other subtypes, comparisons between all subtypes should be descriptively made. Analysis between the predominant subtypes showed no significant difference in OHIP scores; therefore, the null hypothesis H0 cannot be rejected. As recently recommended by John et al., OHRQoL can be divided into four dimensions, all of which are captured by OHIP [20]. In our cohort, “Psychosocial Impact” was lowest, and the “Orofacial Pain” dimension had the highest scores. Our study provides the first data on the four dimensions of OHRQoL in EDS patients and may serve as a reference value for further studies. However, it should be taken into account that the number of participants in individual subtypes was very low.

The mean OHIP value of 19.6 (±12.3) for all subtypes collected in this study is comparable to the results of other studies. For example, Berglund et al. conducted a study in Sweden with 250 EDS participants with an average OHIP score of 11.1 [17]. Oelerich et al. [16] (46 participants) and Hanisch et al. [15] (32 participants) surveyed a median OHIP score of 17 and an average score of 18.9, respectively. All given scores showed a reduced OHRQoL compared to the general population [21]. Compared to a representative sample of the German population from 2003, it was found that only 10% of the general population had an OHIP-49 score above 38 (10.9 points converted to the OHIP-14 score) [22].

To better understand the different OHIP scores and what other OHIP scores mean regarding the quality of life of EDS patients, it is helpful to refer to the study by Reissmann et al., who showed that one point in OHIP-49 is equivalent to 15.2 impacts of impairment of oral health per month [23]. Since we used the short version in this study (OHIP-14), the number of impacts per OHIP point should be multiplied by a factor of 3.5 for comparability, as Reissmann et al. suggested. This means that one point in OHIP-14 corresponds to 53.2 impacts per month, and thus, to approximately two impacts per day. For our study, this means that participants with cEDS, with an OHIP score of 18.0, suffer 957.6 impacts per month (31.9 per day). hEDS participants suffer 1037.4 impacts per month (34.6 per day). The participants with vEDS suffer 808.6 impacts per month (27 per day) if we use the scheme described by Reissmann et al. Many impairments harm the patient’s quality of life and severely limit their daily routine.

Our study shows that the different subtypes differ in their oral symptoms, and their effects on the OHRQoL were highlighted in several areas. In the questionnaire, it is clear that vEDS patients generally do not have as strong oral symptoms compared to patients with the other subtypes. A systematic review has shown that vEDS patients were less frequently affected by dental pathologies, such as caries, pain, and periodontal problems, but were more regularly affected by dentin formation [10]. Studies with more significant participants are needed to investigate whether vEDS patients are subjectively and objectively less affected by oral health-related problems. On the other hand, our data showed a very strong impairment of OHRQoL in participants with BCS and mEDS. Although only a few participants were diagnosed with these subtypes, further studies should address the oral health of BCS and mEDS, as there are no instances in the literature regarding oral health for these subtypes, even though these subtypes showed the worst OHRQoL.

It is also clear that people with subtypes other than vEDS are more affected by dental problems. As an example, people with the periodontal EDS type are suitable. However, since pEDS is characterized by severe periodontitis that begins in early childhood or adolescence, intensive care by the dentist is required in this case. In addition, it is clear that patients experience a decline in OHRQoL [11,24,25].

When asked about the efficacy of local anesthetics, 52.6% of hEDS patients stated that they had experienced a lack of or reduced effect of local anesthetics. This phenomenon has been described in previous studies [26,27]. However, since these responses are subjective perceptions, further studies are needed to clinically and objectively clarify these responses.

Patients with rare diseases must quickly receive a diagnosis, as the occurrence of pain without knowing about the cause can lead to incredible frustration. Patients with a rare disease often suffer for a long time, seeing one physician after another—without any results. Schmitt-Sausen found an average of 7 years between the onset of the first symptoms and diagnosis for patients with rare diseases [28]. Without a diagnosis, patients have no general access to adequate therapies. Our study showed that the 295 participants had to wait an average of 21.8 years (±12.8) for a diagnosis. As mentioned before, Hanisch et al. showed that 56.3% of patients with rare diseases feel poorly supported by the German healthcare system [15]. The long wait for a diagnosis may be closely linked to this. The reason for the significantly longer time between symptoms and diagnosis in EDS patients could be that genetic diagnostic confirmation in EDS patients had not existed for a long time, or not even today, as in the case of the hypermobile subtype. Another reason could be the lack of knowledge among physicians, especially dentists. Kühne et al. showed that the knowledge about rare diseases is significantly higher among dentists working at a university than among non-university dentists [29,30]. Thus, going to university clinics could help shorten the time between the first onset of symptoms and a diagnosis. In a case report on the dental/oral problems in a patient with Ehlers-Danlos syndrome, the authors stated that the consistent dissemination of knowledge about rare diseases among patients, as well as doctors and dentists, is a responsibility for everyone who has even once come into contact with this problem [31].

Bohner et al. showed that a delay in diagnosis leads to a steady decrease in OHRQoL. According to this, in rare diseases with oral symptoms, there is an increase in the OHIP score of 0.08 points with each advancing year without diagnosis [32]. In addition, there is an increased psychological burden in justifying symptoms to others without a current diagnosis. It is, therefore, essential to shorten the time between the appearance of the first symptoms and the diagnosis. Therefore, it is necessary to increase the knowledge of physicians regarding rare diseases such as EDS.

Especially with rare diseases, such as Ehlers-Danlos syndrome, it is difficult to recruit enough participants to conduct studies. As a first step, establishing a national central registry would be helpful to better link research groups and make patient data available to researchers. This is in accordance with the demands of the Commission of the European Communities on rare diseases [2]. Since there are no central patient registries in Germany, we had to rely on the assistance of self-help groups to recruit enough EDS patients for this study. The proportion of women surveyed in this study was 91.5%. This gender imbalance was also described in previous studies, in which men were generally less likely to be members of a self-help group, and the number of women exceeded that of men in some groups by a factor of four to five [33,34,35].

### Limitations

The limiting factor in this study was that not enough participants with rare individual subtypes participated, or subjects of particular subtypes were not represented. Another limitation of this study was the predominance of women at 91.5%. Previous studies have shown similar gender ratios in rare disease studies, where part of the cohort was a support group member [16,36]. However, Bohner et al. showed that gender does not affect OHRQoL in rare diseases [30].

Another limitation of our study was that we could only reach the participants via email distribution lists of the self-help groups and social media groups. EDS patients who did not have internet access or were not active on the internet were, therefore, not included in this study. In addition, the oldest person participating was 65 years old. It might have been possible to recruit more senior participants who were not active on the internet, for example, by displaying flyers at dental practices who offer consultation hours for rare diseases.

All data collected in this study were subjective perceptions of the participants. If the results contained in this study are also objectively measurable, they must be investigated in further studies. Since the questionnaire was distributed solely online, EDS diagnosis was based on self-reporting.

## 5. Conclusions

This study showed that patients with the predominant subtypes cEDS, hEDS, and vEDS had similarly reduced OHRQoL. Noticeable differences between individual EDS subtypes were demonstrated. The time between the onset of first symptoms and diagnosis noticeably differed for predominant subtypes, as participants with vEDS were diagnosed faster than participants with hEDS. It was shown that there was no difference in OHRQoL between EDS patients regardless of whether they were members of a self-help group or not. Subsequent studies are needed to explore if the subjectively perceived reduced OHRQoL can be shown for all subtypes with an adequate number of participants. In addition, further studies are needed to address whether this subjective reduction in OHRQoL manifests itself in different symptoms between the subtypes.

## Figures and Tables

**Figure 1 ijerph-20-02218-f001:**
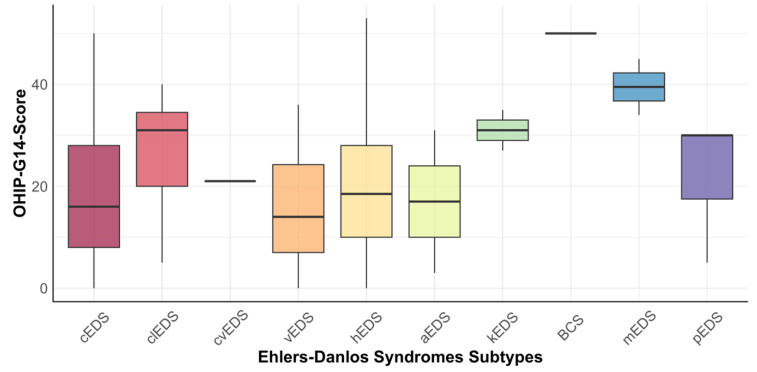
Boxplots for Oral Health Impact Profile-14G across all included subtypes.

**Figure 2 ijerph-20-02218-f002:**
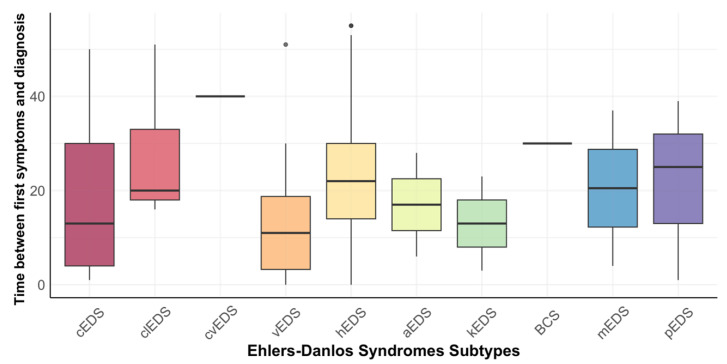
Boxplots for the time between first symptoms and diagnosis in years.

**Figure 3 ijerph-20-02218-f003:**
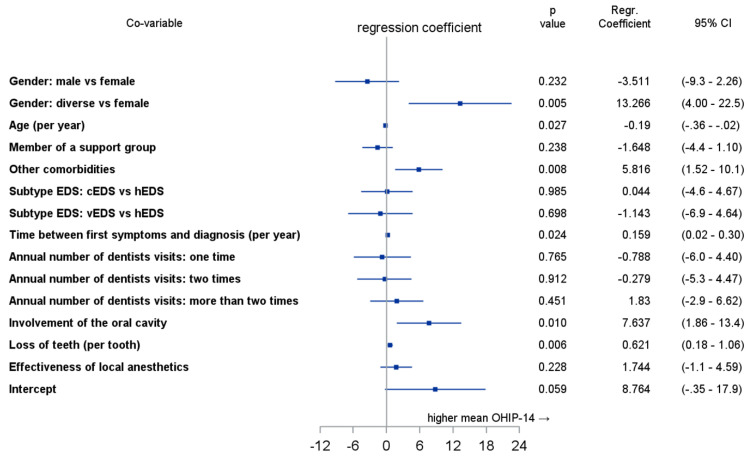
Multivariable linear regression analysis for variables and their impact on Oral Health Impact Profile-14 score.

**Table 1 ijerph-20-02218-t001:** Overview of the results from 3.1 to 3.2; *p*-values for categorical variables were calculated using a Chi-square test, and for continuous variables; a Kruskal-Wallis test was performed to calculate *p*-values; subtype according to the international classification of 2017 [3]; standard deviation not given for cardiac-vascular EDS and Brittle Cornea Syndrome, only one participant, respectively; ^1^ = in years; ^2^ = percentages are given for each subtype.

	Mean (SD)	Range	Classical EDS	Classical-Like EDS	Cardiac-Valvular EDS	Vascular EDS	Hypermobile EDS	Arthrochalasia EDS	Kyphoscoliotic EDS	Brittle Cornea Syndrome	Myopathic EDS	Periodontal EDS	*p*-Value
Count (%)			29 (9.8%)	7 (2.4%)	1 (0.3%)	18 (6.1%)	230 (78.0%)	2 (0.7%)	2 (0.7%)	1 (0.3%)	2 (0.7%)	3 (1.0%)	
Age ^1^	39.2 (±11.2)	18–65	43.0 (±12.4)	45.7 (±11.1)	53	40.1 (±13.0)	38.2 (±10.7)	34,0 (±8.5)	41.5 (±14.8)	63	51.0 (±14.1)	41.7 (± 1.5)	0.129
Gender ^2^													0.465
men			0 (0.0%)	0 (0.0%)	0 (0.0%)	1 (5.6%)	16 (7.0%)	1 (50.0%)	1 (50.0%)	0 (0.0%)	0 (0.0%)	0 (0.0%)	
women			29 (100%)	7 (100.0%)	1 (100.0%)	16 (88.9%)	209 (90.9%)	1 (50.0%)	1 (50.0%)	1 (100.0%)	2 (100.0%)	3 (100.0%)	
diverse			0 (0.0%)	0 (0.0%)	0 (0.0%)	1 (5.6%)	5 (2.2%)	0 (0.0%)	0 (0.0%)	0 (0.0%)	0 (0.0%)	0 (0.0%)	
Country ^2^													0.214
Germany			26 (89.7%)	7 (100.0%)	1 (100.0%)	14 (77.8%)	211 (91.7%)	2 (100.0%)	2 (100.0%)	1 (100.0%)	2 (100.0%)	3 (100.0%)	
Austria			2 (6.9%)	0 (0.0%)	0 (0.0%)	4 (22.2%)	5 (2.2%)	0 (0.0%)	0 (0.0%)	0 (0.0%)	0 (0.0%)	0 (0.0%)	
Switzerland			1 (3.4%)	0 (0.0%)	0 (0.0%)	0 (0.0%)	14 (6.1%)	0 (0.0%)	0 (0.0%)	0 (0.0%)	0 (0.0%)	0 (0.0%)	

**Table 2 ijerph-20-02218-t002:** Calculated scores for the four dimensions of Oral Health-Related Quality of Life (OHRQoL) and Oral Health Impact Profile (OHIP-G14) summary score; each score is given for the individual subtype and as a total across all subtypes; standard deviation not given for cardiac-vascular EDS (cvEDS) and Brittle Cornea Syndrome (BCS), only one participant, respectively.

	Oral Function	Orofacial Pain	Orofacial Appearance	Psychosocial Impact	OHIP-G14 Score
Classical EDS (*n* = 29)	2.41 (±2.40)	3.83 (±2.76)	3.12 (±2.05)	1.86 (±2.08)	18.03 (±12.93)
Classik-like EDS (*n* = 7)	3.29 (±1.98)	6.14 (±2.27)	4.86 (±2.54)	2.86 (±2.48)	26.43 (±12.83)
Cardiac-valvular EDS (*n* = 1)	1.00	2.00	6.00	3.00	21.00
Vascular EDS (*n* = 18)	1.61 (±2.17)	3.17 (±2.43)	2.83 (±2.26)	2.00 (±1.91)	15.17 (±11.56)
Hypermobile EDS (*n* = 230)	2.59 (±2.26)	4.30 (±2.15)	3.70 (±2.31)	1.82 (±1.83)	19.47 (±11.97)
Arthrochalasia EDS (n = 2)	1.00 (±1.41)	2.50 (±2.12)	2.50 (±3.54)	2.50 (±3.54)	17.00 (±19.80)
Kyphoscoliotic EDS (n = 2)	4.50 (±3.54)	6.5 (±2.12)	4.50 (±0.71)	3.00 (±0.00)	31.00 (±5.66)
Brittle Cornea Syndrome (*n* = 1)	6.00	8.00	8.00	6.00	50.00
Myopathic EDS (*n* = 2)	7.50 (±0.71)	7.50 (±0.71)	8.00 (±0.00)	5.00 (±2.83)	39.50 (±7.78)
Periodontal EDS (*n* = 3)	2.00 (±2.65)	4.67 (±1.53)	5.00 (±2.65)	2.67 (±2.52)	21.67 (±14.43)
Total	2.56 (±2.31)	4.26 (±2.28)	3.72 (±2.32)	1.92 (±1.91)	19.56 (±12.29)

**Table 3 ijerph-20-02218-t003:** Calculated mean differences (95% confidence interval) in Oral Health Impact Profile-Scores (OHIP-G14) are shown in the upper half of the table (shaded gray). Corresponding effect sizes (Cohen’s d; 95% confidence interval) are shown in the lower half of the table (not shaded); significant values are bold; Cohen´s d and confidence interval not given for the comparison of cardiac-vascular EDS (cvEDS) and Brittle Cornea Syndrome (BCS), only one participant, respectively; n.a. = not applicable.

	cEDS	clEDS	cvEDS	vEDS	hEDS	aEDS	kEDS	BCS	mEDS	pEDS
cEDS	–	8.39 (−2.65–19.44)	2.97 (−23.97–29.90)	−2.87 (−10.38–4.64)	1.44 (−3.25–6.12)	−1.03 (−20.81–18.74)	12.97 (−6.09–32.02)	**31.97 (5.03–58.90)**	**21.47 (2.35–40.58)**	3.63 (−12.51–19.78)
clEDS	−0.65 (−1.49–0.19)	–	−5.43 (−38.99–28.13)	**−11.26 (−22.23–−0.30)**	−6.96 (−16.02–2.10)	−9.43 (−36.05–17.19)	4.57 (−18.31–27.45)	23.57 (−9.99–57.13)	13.07 (−10.13–36.27)	−4.76 (−25.85–16.32)
cvEDS	−0.23 (−2.22–1.77)	0.42 (−1.70–2.51)	–	−5.83 (−30.88–19.22)	−1.53 (−25.16–22.10)	−4.00 (−312.11–304.11)	−10.00 (−78.03–98.03)	29.00	18.50 (−102.54–139.54)	0.67 (−71.04–72.39)
vEDS	0.23 (−0.36–0.82)	**0.95 (0.02–1.85)**	0.51 (−1.52–2.52)	–	4.30 (−1.45–10.06)	1.83 (−17.21–20.88)	15.83 (−1.88–33.54)	**34.83 (9.78–59.88)**	**24.33 (6.51–42.15)**	6.50 (−9.02–22.02)
hEDS	−0.12 (−0.51–0.27)	0.58 (−0.17–1.33)	0.13 (−1.84–2.09)	−0.36 (−0.84–0.12)	–	−2.47 (−19.28–14.34)	11.53 (−5.19–28.25)	**30.53 (5.90–54.16)**	**20.03 (3.31–36.76)**	−2.20 (−11.53–15.93)
aEDS	0.08 (−1.36–1.51)	0.67 (−0.96–2.26)	0.20 (−2.25–2.57)	−0.15 (−1.61–1.31)	0.21 (−1.19–1.60)	–	14.00 (−48.65–76.65)	33.00 (−275.11–341.11)	22.50 (−42.22–87.22)	4.67 (−43.03–52.36)
kEDS	−1.02 (−2.47–0.45)	−0.38 (−1.95–1.22)	−1.77 (−4.82–1.56)	−1.40 (−2.91–0.15)	−0.97 (−2.36–0.43)	−0.96 (−3.02–1.27)	–	19.00 (−69.03–107.03)	8.50 (−20.76–37.76)	−9.33 (−44.86–26.20)
BCS	**−2.47 (−4.55–−0.36)**	−1.84 (−4.11–0.55)	n.a.	**−3.01 (−5.23–−0.73)**	**−2.55 (−4.53–−0.57)**	−1.67 (−4.64–1.59)	−3.36 (−8.01–1.19)	–	−10.50 (−131.54–110.54)	−28.33 (−100.04–43.38)
mEDS	**−1.68 (−3.16–−0.17)**	−1.07 (−2.70–0.63)	−2.38 (−6.00–1.40)	**−2.14 (−3.73–−0.50)**	**−1.68 (−3.07–−0.27)**	−1.50 (−3.78–0.99)	−1.25 (−3.42–1.11)	1.35 (−1.69–4.08)	–	−17.83 (−54.47–18.81)
pEDS	−0.28 (−1.47–0.91)	0.36 (−1.02–1.71)	0.05 (−2.30–2.22)	−0.55 (−1.77–0.70)	−0.18 (−1.32–0.96)	−0.28 (−2.07–1.54)	0.76 (−1.17–2.59)	1.96 (−1.03–4.73)	1.41 (−0.75–3.43)	–

**Table 4 ijerph-20-02218-t004:** Overview of the results from 3.5; *p*-values for categorical variables were calculated using a Chi-square test, and for continuous variables, a Kruskal-Wallis test was performed to calculate *p*-values; ^1^ = in years.

	Classical EDS	Classical-Like EDS	Cardiac-Valvular EDS	Vascular EDS	Hypermobile EDS	Arthrochalasia EDS	Kyphoscoliotic EDS	Brittle Cornea Syndrome	Myopathic EDS	Periodontal EDS	*p*-Value
**Time of diagnosis ^1^**	27.7 (±18.5)	43.1 (±12.2)	43	33.6 (±12.6)	34.9 (10.8)	31.5 (±7.8)	24.5 (±6.4)	58	48.0 (±15.6)	27.3 (±21.1)	0.086
**Time between first symptoms and diagnosis ^1^**	19.8 (±17.2)	27.0 (±13.3)	40	13.3 (±13.0)	22.5 (±11.8)	17.0. (±15.6))	13.0 (±14.1)	30	20.5 (±23.3)	21.7 (±19.2)	0.058
**Involvement of the oral cavity**											0.004
yes	24 (82.8%)	6 (85.7%)	1 (100.0%)	13 (72.2%)	222 (96.5%)	2 (100.0%)	2 (100.0%)	1 (100.0%)	2 (100.0%)	3 (100.0%)	
no	5 (17.2%)	1 (14.3%)	0 (0.0%)	5 (27.8%)	8 (3.5%)	0 (0.0%)	0 (0.0%)	0 (0.0%)	0 (0.0%)	0 (0.0%)	

## Data Availability

All data sets are deposited at the Clinic for Oral and Maxillofacial Surgery of the University Hospital Münster and can be accessed.

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
