# Peer review of "Oral Health-Related Quality of Life in Different Subtypes of Ehlers-Danlos Syndrome"

_ijerph, 2023, doi:10.3390/ijerph20032218_

Round 1
Reviewer 1 Report
This aim of the study was to assess differences in the oral health-related quality of life (OHRQoL) between subtypes of Ehlers-Danlos syndromes (EDS).
In the Introduction the Authors provided some general information about the Ehlers-Danlos syndrome (EDS) and its classification. They also underlined the importance of the research. The aim of the study has been precisely defined.
The methodology applied has been discussed in sufficient details. The statistical analyses used in the study are adequate.
The results are quite clearly discussed. However, Tables 1 and 3 are illegible and should be corrected.
In the Discussion the Authors discussed the obtained results and compared with other findings. The limitations of the study were also discussed.
Author Response
We would like to thank you for all of your remarks. We tried to implement all changes suggested by you as follows: The results are quite clearly discussed. However, Tables 1 and 3 are illegible and should be corrected. We revised Tables 1 and 3 (now Tables 1 and 4 due to other changes). We think they are better readable now.Reviewer 2 Report
The submitted paper is significant as it analyzes the different types of Ehlers Danlos Syndrome (EDS) and oral health related quality of life.
Introduction:
"Fewer than five in 10,000 people": dot to be replaced with a comma
In the sentence, .........."other hand, have lower diagnostic specificity and have a supporting character in the diagnosis": it must be characteristics instead of character
"Due to mutations in the affected genes, defects occur in fibrillar collagen types I, III, 54 and IV". Reference needed
"Due to the presence of collagen in all tissues, it is understandable that oral symptoms also occur". Not clear. It would be ideal to relate it with EDS and rewrite the sentence.
"The somewhat lower power used here was justified by the fact that the population under investigation is a rare disease, which makes it 90 challenging to recruit patients". What will be the sample size for a power of 80%?
Were participants contacted through email?
How many methods were used to recruit participants?
Was there any database used to identify patients with EDS?
Is recruitment based on self-reporting of EDS?
Was the survey administered in both German and English languages?
What were the reasons to exclude cases? Exclusion criteria are not provided.
Table 2 questions are incomplete. Is the dots …. seen in table 2 questions intentional or error in formatting?
Figure 1 axis titles could be expanded (Example: Ehlers-Danlos syndrome subtypes), and the axis lines could be added by removing the gridlines.
Are there any reason for not comparing the oral health related quality of life with the general population?
Author Response
First of all, we would like to thank you for your comments. We implemented the following changes according to your remarks: "Fewer than five in 10,000 people": dot to be replaced with a comma We replaced the dot with a comma in line 34.In the sentence, .........."other hand, have lower diagnostic specificity and have a supporting character in the diagnosis": it must be characteristics instead of character
We changed character to characteristics in line 41.
"Due to mutations in the affected genes, defects occur in fibrillar collagen types I, III, 54 and IV". Reference needed
We added the Reference in line 54. Thank you for noticing.
"Due to the presence of collagen in all tissues, it is understandable that oral symptoms also occur". Not clear. It would be ideal to relate it with EDS and rewrite the sentence.
This sentence was rewritten for better understanding in lines 58-59.
"The somewhat lower power used here was justified by the fact that the population under investigation is a rare disease, which makes it 90 challenging to recruit patients". What will be the sample size for a power of 80%?
The sample size required for a power of 80% is 336 participants. We received a positive vote from the Ethics commitee based on our reduced power of 76.3%. This was justified as stated in the manuscript by the fact that the population under investigation is a rare disease.
Were participants contacted through email?
We specified this passage in lines 85-86. Self-support groups were contacted by mail and informed their members both by mail and via their website.
How many methods were used to recruit participants?
As mentioned in Section 2.1 (lines 78-93) we used three methods to recruit participants:
- Self-support groups were contacted to inform their members
- Social media Groups (Facebook) were used to inform affected individuals with EDS
- Researchers working on the field of EDS were contaced to inform their patients
Was there any database used to identify patients with EDS?
There were no databases used to identifying patients with EDS as there are no such databases currently existing in Germany. We mentioned the need for such databases/registries to be established in lines 375-376.
Is recruitment based on self-reporting of EDS?
Recruitment was based on self-reporting. We added this to our limitations in lines 400-401. Participants reported being diagnosed based on the current classification.
Was the survey administered in both German and English languages?
The survey was solely administered in German language as stated in lines 75-76 in three German-speaking countries Germany,Austria and Switzerland.
What were the reasons to exclude cases? Exclusion criteria are not provided.
Reasons for exclusion of cases was added in lines 170-173. Thank you for noticing.
Table 2 questions are incomplete. Is the dots …. seen in table 2 questions intentional or error in formatting?
Former Table 2 is now Supplementary Table 1 due to other changes. For clarity we have changed the table. It consists of three sentence beginnings, which are now written in bold and end with "...". They are followed by the corresponding sentence endings beginning with "...". This was done in order not to write out each question completely and to shorten the table. We hope that this explains your comment.
Figure 1 axis titles could be expanded (Example: Ehlers-Danlos syndrome subtypes), and the axis lines could be added by removing the gridlines.
We changed Figure 1 and 2 according to your recommendations. After careful consideration we decided to leave the gridlines remaining for better readability.
Are there any reason for not comparing the oral health related quality of life with the general population?
We made a comparison of OHIP-14 scores between our cohort and the general population in lines 309-312. Please let us know if you think further comparision is needed.
Again we would like to thank you for all of your remarks on this manuscript.
Reviewer 3 Report
The manuscript “Oral health-related quality of life in different subtypes of Ehlers-Danlos syndromes” aimed “to assess differences in the oral health-related quality of life (OHRQoL) 15 between subtypes of Ehlers-Danlos syndromes (EDS).”
The topic is interesting, relevant, and new for IJERPH readers.
I have few suggestions to improve the manuscript.
I do not think that hypothesis testing is a good tool to investigate differences between different subtypes of Ehlers-Danlos syndromes. This approach leads to statements such as “No significant difference in OHIP score between the ten subtypes participating in the 184 study could be shown (p = 0.116).” Such a statement is not informative because statistical significance is not related to clinical relevant. In small samples, relevant (large) differences can be missed. In large samples, not relevant (small) differences could be statistically significant.
Instead, in the main analysis, the authors should present the differences between subtypes as a difference in means (or medians) that is accompanied by a confidence interval. Such results could be presented in a matrix with the subtypes as columns and rows (only one triangle of the matrix needs to be presented). The author should also describe the differences with effect sizes (also accompanied by confidence intervals) to give the reader information about the magnitude of the differences. Cohen’s guidelines for the magnitude of differences should be used.
While OHIP-14 is a solid instrument, the authors need to consult “Recommendations for use and scoring of Oral Health Impact Versions” and their validation (both articles: John et al J Evid Based Dental Practice 2022). The entire analysis does not need to be repeated for each of the four dimensions of OHRQoL, but for each subtype the 4-dimensional score profile should be presented to describe the dimensional OHRQoL impact. Using this information allows to compared the study with already published or future studies showing OHRQoL dimensional profiles.
Author Response
First of all we want to thank you for your remarks on this manuscript. We believe that these comments, in particular, significantly improve the statistical significance of our work.
We undertook significant changes in our main analysis and made a comparison of the means of each subtype with associated effect sizes (Cohen´s d was used). Confidence intervals (95%) were also reported. As noted by you, this presentation in Table 3 allows a direct comparison between subtypes to better illuminate potential differences between them. We have marked any noticeable differences in bold, both for mean differences and effect sizes.
Since we planned and conducted this study based on our hypothesis, we continue to think that it should nevertheless be mentioned in addition. We think that the added data will help to support this hypothesis with sufficient information and show informative character, because now a much better comparison between subtypes can be made.
During the study planning, the "Recommendations for use and scoring of Oral Health Impact Versions" did not yet exist and were therefore not considered. Thanks to your comments, we were able to implement these rather new recommendations and, like you, we think that this will contribute enormously to making the results comparable in the future. We have calculated the four dimensions of the OHRQoL in Table 2 for each subtype and added a supplement in section 2.4 (lines 117-136). Furthermore, we have also tested our hypothesis for the four dimensions and could show that for the dimension "Orofacial Pain" a noticeable difference between the subtypes could be shown. In addition, we added effect sizes for all four dimensions for individual comparison of the subtypes in Supplementary Table 4.
We would like to thank you again for your comments as we think they have significantly improved the quality of our work.
Round 2
Reviewer 2 Report
Dear authors,
Thank you for addressing the comments. Your initiative to understand the oral health related quality of a rare connective tissue disorder such as Ehlers-Danlos syndrome and its different types is much appreciated.
Reviewer 3 Report
Thank you for the constructive response. I have no further comments.